# Estimation of above- and below-ground ecosystem parameters for the DVM-DOS-TEM v0.7.0 model using MADS v1.7.3

Elchin E. Jafarov[1], Hélène Genet[2], Velimir V. Vesselinov (Monty)[3], Valeria Briones[1], Aiza Kabeer[4], Andrew L. Mullen[1], Benjamin Maglio[2], Tobey Carman[2], Ruth Rutter[2], Joy Clein[2], Chu-Chun Chang[1], Dogukan Teber[1], Trevor Smith[1], Joshua M. Rady[1], Christina Schädel[1], Jennifer D. Watts[1], Brendan M. Rogers[1], Susan M. Natali[1]

[1]Woodwell Climate Research Center, Falmouth, MA, USA

[2]Institute of Arctic Biology, University of Alaska Fairbanks, Fairbanks, AK, USA

[3]EnviTrace LLC, NM, USA

[4]Program in Applied Mathematics, University of Arizona, Tucson, AZ, USA

*Correspondence to*: Elchin E. Jafarov (ejafarov@woodwellclimate.org)

**Abstract.**

The permafrost region contains a significant portion of the world's soil organic carbon, and its thawing, driven by accelerated Arctic warming, could lead to the substantial release of greenhouse gases, potentially disrupting the global climate system. Accurate predictions of carbon cycling in permafrost ecosystems hinge on the robust calibration of model parameters. However, manually calibrating numerous parameters in complex process-based models is labor-intensive and further complicated by equifinality - the presence of multiple parameter sets that can equally fit the observed data. Incorrect calibration can lead to unrealistic ecological predictions. In this study, we employed the Model Analysis and Decision Support (MADS) software package to automate and enhance the accuracy of parameter calibration for carbon dynamics within the coupled Dynamic Vegetation Model, Dynamic Organic Soil Model, and Terrestrial Ecosystem Model (DVM-DOS-TEM), a process-based ecosystem model designed for high-latitude regions. The calibration process involved adjusting rate-limiting parameters to accurately replicate observed carbon and nitrogen fluxes and stocks in both soil and vegetation. Gross primary production, net primary production, vegetation carbon, vegetation nitrogen, and soil carbon and nitrogen pools served as synthetic observations for a black spruce boreal forest ecosystem. To validate the efficiency of this new calibration method, we utilized model-generated synthetic and actual observations. When matching model outputs to observed data, we encountered difficulties in maintaining mineral soil carbon stocks. Additionally, due to strong interdependencies between parameters and target values, the model consistently overestimated carbon and nitrogen allocation to the stem of evergreen tree. This study demonstrates the calibration workflow, offers an in-depth analysis of the relationships between parameters and observations (synthetic and actual), and evaluates the accuracy of the calibrated parameter values.

## 1 Introduction

The permafrost region contains 1,440-1,600 petagrams of organic carbon in its soils, representing nearly half of the world's soil organic carbon pool (Hugelius et al., 2014; Schuur et al., 2022). Accelerated warming in the Arctic leads to permafrost thaw, resulting in the decomposition and potential release of a substantial portion of this stored carbon as greenhouse gases, significantly impacting the global climate system (Natali et al., 2021; Schuur et al., 2022; Treharne et al., 2022). The permafrost carbon-climate feedback remains one of the largest sources of model uncertainty for future climate predictions, as critical ecological and biogeochemical processes are poorly represented and constrained in ecosystem models, if included at all (McGuire et al., 2016, 2018; Schädel et al., 2024). A significant portion of this uncertainty stems from parameter uncertainty, particularly in rate-limiting factors that control biogeochemical cycles, which are challenging to measure directly and can vary considerably across spatial and temporal scales (Koven et al., 2015; Mishra et al., 2021). These uncertainties propagate through model simulations, contributing to a wide range of projected permafrost carbon emissions (Lawrence et al., 2015; McGuire et al., 2018).

When compared to structural uncertainty (which arises from incomplete or simplified representations of ecological processes) and input data uncertainty (resulting from limited or biased forcing datasets), parameter uncertainty is particularly pervasive and difficult to constrain (Euskirchen et al., 2022; Fisher and Koven, 2020; Luo et al., 2016). While structural uncertainties limit a model's ability to fully capture real-world processes, parameter uncertainties directly alter numerical outputs, often amplifying variations in projections (Fisher and Koven, 2020; Turetsky et al., 2020). Models are particularly sensitive to parameter uncertainties, given the complexity and variability of the processes they simulate, including soil thermal dynamics, vegetation feedbacks, and hydrological interactions (Andresen et al., 2020; Harp et al., 2016; Koven et al., 2015). While structural improvements to model frameworks are ongoing, addressing parameter uncertainty through robust calibration methods remains an essential and complementary step for enhancing the accuracy and reliability of model outputs (Fisher and Koven, 2020; Luo et al., 2016). Addressing these uncertainties through the development of effective calibration techniques is essential for refining predictions of permafrost dynamics and better constraining future permafrost carbon-climate feedbacks (McGuire et al., 2018; Mishra et al., 2021).

Calibration involves estimating and adjusting model parameters to enhance the agreement between model outputs and observed data, with the model serving as a mathematical representation of ecological and physical processes (Rykiel, 1996). These parameters are often rate or transport constants that are onerous or impractical to empirically estimate, though model outputs can be highly sensitive to them. Since many model representations are grounded in physics, generalized physical laws are often used to describe ecological and cryohydrological processes. Typically, model outputs are validated against data from laboratory experiments, idealized mathematical models, or site-specific observations, also referred to as target data. During this validation, model parameters are adjusted so that model outputs match the target data. The validated model is then applied to broader geographic locations and/or different time periods, assuming that the validation data represent the environment or ecosystem for which the parameters were calibrated.

Parameter calibration for complex process-based models is often constrained by the significant labor required and the limited availability of sites with the necessary observations, especially in permafrost regions (Birch et al., 2021; Virkkala et al., 2019). Despite these challenges, process-based models remain essential because they encapsulate our current understanding of ecosystem functions and structures, serving as powerful tools for extrapolation. The assumption of representativeness is intrinsic to these models, as they are designed to simulate processes that reflect our best understanding of ecosystem dynamics, allowing for their application beyond the individual sites where they have been initially parameterized. The approach of extrapolating model parameterization for ecosystems of the same type, across wider regions is standard and widely used within ecosystem modeling communities (Matthes et al., 2025; McGuire et al., 2018). Additionally, the role of ecosystem diversity on the spatio-temporal patterns of ecosystem carbon dynamics in the permafrost region has been characterized by numerous empirical studies (Euskirchen et al., 2014; Melvin et al., 2015) and evaluated by modeling investigations (Lara et al., 2016). Therefore, a critical step in improving model accuracy involves calibrating the model against a suite of data for a representative diversity of ecosystem types in the Arctic where observations are available. To prepare an ecosystem model for this extensive calibration task, it is essential to develop robust calibration tools and methods that can automate the process of efficiently optimizing model parameters.

Another well-known and significant issue in optimizing model parameters through calibration, also referred to as parameter estimation or optimization, is the existence of equifinality (Jafarov et al., 2020; Nicolsky et al., 2007; Tran et al., 2017). Parameterization equifinality occurs when different sets of parameter values result in the same or similar model predictions, given that the model, forcing data, and observations used in calibration are the same (Beven and Freer, 2001). Model equifinality can subsequently lead to different outcomes in model projections. In an aim to address the issue of equifinality, we run the model using randomly varied parameter values within the given range. If the majority of calibration tests with different initial guesses yield a good fit with observations and result in optimal parameter sets that are similar or closely aligned, it increases confidence that the recovered parameter set is indeed optimal. This approach mitigates the risk of converging on a local minimum and ensures a more robust and reliable parameter estimation process (Hansen, 1998).

Various methods have been employed to improve the calibration of model parameters across multiple scientific disciplines, utilizing sophisticated techniques and integrating diverse data sources such as remote sensing and field measurements, while accounting for model and data uncertainty (Dietze et al., 2018; Efstratiadis and Koutsoyiannis, 2010; Luo et al., 2016). Optimization-based inverse methods have been successfully used to calibrate parameters in physical models, including snow properties and subsurface thermo-hydrological properties (Jafarov et al., 2014, 2020), as well as soil properties for permafrost modeling (Nicolsky et al., 2007, 2009). However, inverse modeling can become computationally intractable when applied to complex process-based models (Linde et al., 2015).

Markov Chain Monte Carlo (MCMC) and data assimilation (DA) techniques have been employed to optimize model parameters by synchronizing model outputs with observed data, thereby enhancing model prediction accuracy (Brunetti et al., 2023; Fer et al., 2018; Xu et al., 2017). These methods often

leverage Bayesian inference to address structural uncertainties within models. Nonetheless, the computational demand required for conducting MCMC simulations can outweigh the gains in model accuracy, particularly when dealing with complex process-based models with slow turnover rates that necessitate long simulations to reach equilibrium.

In recent years, DA techniques have been applied to optimize both model state variables (Fox et al., 2018; Ling et al., 2019) and parameters (Bloom et al., 2016; Peylin et al., 2016; Scholze et al., 2016; Schürmann et al., 2016). However, DA also encounters challenges related to unbalanced outputs and the need for extended simulations to achieve equilibrium. Persistent issues include the incorrect characterization of the error covariance matrix, which can lead to inaccurate posterior parameter values due to unaccounted model structural errors and observation biases (MacBean et al., 2016; Wutzler and Carvalhais, 2014).

Various surrogate-based optimization approaches have been proposed to alleviate the computational burden associated with parameter calibration (Koziel et al., 2011; Queipo et al., 2005). Surrogate models, also known as reduced-order models, simplify certain physical processes to approximate the underlying dynamics of the real model while being computationally less demanding (Forrester et al., 2006). By simplifying specific aspects of the model, surrogate models retain essential characteristics of the original system, allowing for faster and more efficient calibration without significantly compromising accuracy (Razavi et al., 2012; Regis and Shoemaker, 2007). However, simplifying complex models presents significant challenges. It is often unclear which assumptions can be safely made and which should be avoided, potentially leading to a loss of model accuracy. Surrogate models must carefully balance the trade-off between simplification and the retention of critical model characteristics to ensure reliable performance. This complexity necessitates rigorous validation to confirm that the surrogate model provides an adequate approximation of the real system without introducing significant errors.

In recent years, machine learning-based emulators, often referred to as "models of models," have emerged as a promising approach to reduce the computational burden associated with parameter calibration in complex ecosystem models (Castelletti et al., 2012; Fer et al., 2018; Reichstein et al., 2019). These emulators aim to approximate the outputs of physical and process-based models by learning the relationships between model inputs and outputs through multi-dimensional matrices, significantly enhancing computational efficiency. Unlike traditional surrogate models, which simplify the physical processes within a model, emulators strive to mimic the full complexity of the original model while requiring less computational power. For instance, Dagon et al., (2020) utilized artificial neural networks to emulate the Community Land Model version 5 outputs, focusing on biophysical parameter estimation and global calibration. By integrating machine learning techniques, they were able to explore parameter spaces more efficiently and achieve better alignment with observed data. This method demonstrates the potential of machine learning emulators in improving the accuracy and efficiency of parameter calibration in ecosystem models, particularly when faced with the challenge of high computational demands.

To facilitate the automation of the calibration process while minimizing computational demand and avoiding the oversimplification of ecological processes and feedbacks, we employed a non-linear least squares approach for our calibration. We utilized the Model Analysis and Decision Support (MADS) software package (Barajas-Solano et al., 2015; O'Malley and Vesselinov, 2015) for parameter calibration of a terrestrial ecosystem permafrost-enabled model. MADS has been actively developed since 2010, and its conversion to the Julia programming language has provided automatic differentiation capabilities suitable for calibration problems, improving computational efficiency (Vesselinov V.V., 2022).

In this study, we developed an automated parameter calibration method for a process-based terrestrial ecosystem model developed for high-latitude regions and characterized by a high level of complexity. To demonstrate its efficacy, we utilized synthetic data and evaluated the capacity of the calibration method to recover the data after perturbing initial guesses (a given set of parameters) using random sampling. The model was run using known parameter values, and the resulting outputs were treated as observations. The primary objective was to illustrate that the parameter calibration method could recover the synthetic parameter set successfully. The secondary objective was to optimize and reduce the labor and time associated with manual parameter calibration. We developed and tested our calibration method for the coupled dynamic vegetation model, dynamic organic soil, and terrestrial ecosystem model (DVM-DOS-TEM) and tested our approach using synthetic and site observations at a black spruce forest site, a dominant community type in Interior Alaska.

## 2 Methods

### 2.1 Black Spruce Forest site

Approximately 39% of Interior Alaska is covered by evergreen forest stands, dominated by white or black spruce and 24% by deciduous forest stands, dominated by Alaska paper birch or trembling aspen (Calef et al., 2005; Jean et al., 2020). In our study, we developed model calibration for a black spruce (*Picea mariana*) forest community type, using observations collected in a site located within the Tanana Valley State Forest, just outside Fairbanks, Alaska (64°53′N, 148°23′W). Carbon (C) and nitrogen (N) cycling and environmental monitoring in this forest stand were originally observed by Melvin et al., (2015). The stand resulted from a self-replacement succession trajectory following the 1958 Murphy Dome fire, which covered 8,930 hectares.

### 2.2 DVM-DOS-TEM description

DVM-DOS-TEM is a process-based biosphere model designed to simulate biophysical and biogeochemical processes between the soil, vegetation, and atmosphere. DVM-DOS-TEM has been applied extensively in Arctic and Boreal ecosystems in permafrost and non-permafrost regions (Briones et al., 2024; Euskirchen et al., 2022; Genet et al., 2013, 2018; Jafarov et al., 2013; Yi et al., 2009, 2010). This model focuses on representing C and N cycles in high-latitude ecosystems and how they are affected at seasonal (i.e., monthly) to centennial scales by climate, disturbances (Genet et al., 2013,

2018; Kelly et al., 2013), biophysical processes such as soil thermal and hydrological dynamics (McGuire et al., 2018; Yi et al., 2009; Zhuang et al., 2002), snow cover (Euskirchen et al., 2006), and plant canopy development (Euskirchen et al., 2014). Modeled vegetation is structured into multiple tiers: (1) the community type (CMT) represents the land cover class and characterizes vegetation composition and soil structure at the gridcell level (spatial unit, e.g. black spruce forest, tussock tundra, bog), (2) plant functional types (groups of species sharing similar functional traits) characterize the vegetation composition within every CMT (e.g. black spruce forest community would be composed of evergreen trees, deciduous shrubs and sphagnum and feather moss plant functional types), and (3) plant structural compartments (leaves, stems, roots). The soil column is split into multiple horizons (fibric, humic, mineral, and rock/parent material). Every horizon is split into multiple layers for which C, N, temperature, and water content are simulated individually. The biophysical processes represented in DVM-DOS-TEM include radiation and water fluxes between the atmosphere, vegetation, snow cover, and soil column. Soil moisture and temperature are updated at a pseudo-daily time step (from linear interpolation of monthly climate forcings). A two-directional Stefan Algorithm is used to predict the positions of freezing/thawing fronts in the soil. The Richards equation is used to calculate soil moisture changes in the unfrozen layers of soil. Both the thermal and hydraulic properties of soil layers are affected by their water content (Yi et al., 2009, 2010; Zhuang et al., 2002). The ecological processes represented in DVM-DOS-TEM include C and N dynamics for every plant functional type (PFT) of the vegetation community and every layer of the soil column. $C$ and $N$ dynamics are driven by climate, atmospheric $CO_2$ content, soil and canopy environment, and wildfire occurrence and severity. $C$ and $N$ cycles are coupled in the soil and the vegetation processes. The growth primary productivity (GPP) of each PFT  is limited by $N$ availability. When resources in N are limited, GPP is downregulated for all PFTs based on a comparison of $N$ demand (N required to build new tissues) and N supply in the ecosystem (Euskirchen et al., 2009). $C$ and $N$ from the litterfall are divided into aboveground and belowground. Aboveground litterfall is assigned only to the top layer of the soil column, while belowground litterfall (root mortality) is assigned to different layers of the three soil horizons based on the fractional distribution of fine roots with depth.

**2.3 Synthetic data**
We used GPP without N limitation (GPP*), Net Primary Productivity (NPP), Vegetation C, and Vegetation N stocks by compartments (i.e. roots, stems, and leaves) as synthetic observations shown in Table 1. Synthetic observations are model-generated data that simulate actual measurements using known parameter values, referred to as synthetic target values. To generate these target values, we used existing parameters and the setup described in Section 2.3. The target values shown in Table 1 represent the state of the ecosystem where vegetation and below-ground C stocks are in a steady state. Table 2 includes the below-ground target values. The model was previously manually calibrated using observations from the site. The actual observations were collected and prepared from the measured data at the site and from existing literature and published datasets. Data pre-processing was required before the time series data could be analyzed. Pre-processing was performed to identify and resolve missing data, inconsistencies, and potential outliers. In addition, site observations were aggregated to a monthly resolution to match the temporal resolution of the model outputs, and unit transformations were applied when needed to standardize the units of each variable. Target values for the site were compiled from

various data literature sources containing information on C and N stocks, plant biomass, soil horizon
depths, and productivity. However, following the initial calibration, the model outputs were similar but
did not exactly match the target observations. As stated above, we choose synthetic targets because we
know a set of parameters used to produce them and can compare how closely we can recover known
parameter values. Therefore, we used the actual model output as our synthetic target values.
**Table 1**: Synthetic vegetation target values for the black spruce forest site used in the parameter
calibration process

| Above-ground Target Names | Notation | Units | Plant Functional Types | | | |
|---|---|---|---|---|---|---|
| | | | Evergreen Tree | Deciduous Shrub | Deciduous Tree | Moss |
| Gross Primary Productivity without nitrogen limitation | $GPP^*$ | [gC/m²/year] | 307.17 | 24.53 | 46.53 | 54.23 |
| Net Primary Productivity | $NPP$ | [gC/m²/year] | 113.08 | 11.3 | 24.02 | 32.41 |
| Vegetation Carbon Leaf | $C_{leaf}$ | [gC/m²] | 572.36 | 8.35 | 6.14 | 136.54 |
| Vegetation Carbon Stem | $C_{stem}$ | [gC/m²] | 1894.03 | 98.90 | 477.80 | |
| Vegetation Carbon Root | $C_{root}$ | [gC/m²] | 474.55 | 33.19 | 7.17 | |
| Vegetation Nitrogen Leaf | $N_{leaf}$ | [gC/m²] | 14.79 | 0.38 | 0.57 | 1.15 |
| Vegetation Nitrogen Stem | $N_{stem}$ | [gC/m²] | 30.26 | 2.6 | 12.53 | |
| Vegetation Nitrogen Root | $N_{root}$ | [gC/m²] | 9.51 | 0.72 | 0.16 | |

**Table 2**: Synthetic below-ground target values for the black spruce forest site used in the parameter
calibration process

| Below-ground Targets Names | Notation | Unit | Value |
|---|---|---|---|
| Carbon Shallow | $C_{shallow}$ | g/m2 | 888.91 |

| Carbon Deep | $C_{deep}$ | g/m2 | 3174.53 |
|---|---|---|---|
| Carbon Mineral Sum | $\sum C_{mineral}$ | g/m2 | 19821.50 |
| Available Nitrogen Sum | $\sum N_{avail}$ | g/m2 | 0.76 |


## 2.4 Input data used for equilibrium run

The driving inputs for the DVM-DOS-TEM model comprise spatial distribution of CMTs, landform,
and mineral soil texture. These initialization data were forced to field observations at the study site
(Melvin et al., 2015). The spatiotemporal dynamics of the model are driven by an annual time series of
atmospheric $CO_2$ concentration (not spatially explicit), annual time series of spatially explicit
distribution of fire scars and dates, and a spatially explicit monthly time series of climate, including
mean air temperature, total precipitation, net incoming shortwave radiation, and vapor pressure (Genet
et al., 2018). For the present study, we use historical climate data from 1901 to 2015, sourced from the
Climatic Research Unit time series version 4.0 (CRU TS4.0; Harris et al., 2014) and downscaled at a 1-
km resolution using the delta method (Pastick et al., 2017). For the equilibrium run, the model was
driven using the averaged climate forcings from the 1901-1930 period for the study site location,
repeated continuously for a sufficient period so equilibrium of vegetation and below-ground C and N
fluxes and stocks was achieved. The resulting modeled ecosystem state for each site is then used to
initialize historical simulations. However, the calibration process described here only utilized outputs
from the equilibrium.

## 2.5 MADS parameter calibration

We employed the MADS software package for parameter calibration of DVM-DOS-TEM, aiming to
minimize the discrepancy between synthetic target and modeled data at the selected site (Barajas-Solano
et al., 2015; O'Malley and Vesselinov, 2015). Since its inception in 2010, MADS has undergone active
development, including a transition to the Julia programming language, which supports automatic
differentiation suitable for calibration problems(Vesselinov V.V., 2022).
The MADS package utilizes the Levenberg-Marquardt (LM) algorithm (Levenberg, 1944; Marquardt,
1963; Pujol, 2007) to minimize the difference (the sum of squared residuals) between observations and
modeled predictions. In SI1, we provide more details on the LM algorithm. The LM optimization
method designed to solve non-linear least squares optimization/minimization problems, which are
common in the field of history matching, model inversion, curve fitting, and parameter estimation. It
combines two approaches: the first-order steepest-descent gradient method and the second-order Gauss-
Newton method. This steepest-descent gradient method updates parameter values in the direction
opposite to the gradient, thereby it is generally efficient in finding local minima. The Gauss-Newton

method assumes that in a region close to the solution, the solved objective function behaves quadratically.

The algorithm begins by selecting an initial estimate for the parameters that need to be optimized (Fig S1). This initial guess is important as it sets the starting point for the optimization process. In our experiment, the initial guess is randomly generated from within the provided range near `true` parameter values. Alternatively, users can provide the initial guess. However, exploring a set of random initial guesses provides an efficient approach to exploring the parameter space and discrimination between local and global minima. In LM, we set the damping parameter (the Marquardt lambda) to 0.01. This parameter helps in adjusting the steps taken during the optimization process, balancing between the two optimization strategies (the first- and the second order techniques discussed above). The main advantages of the LM method are its robustness and minimal computational demand. It effectively handles ill-conditioned problems where other optimization methods might fail (Lin et al., 2016; Pujol, 2007). Additionally, for problems well-suited to the Gauss-Newton method, LM often converges faster than gradient descent, making it an efficient choice for many non-linear least squares problems.

The disadvantage of the LM method is its sensitivity to the initial parameter guesses, potentially affecting its efficiency and convergence (Transtrum and Sethna, 2012). In these cases, MADS provides alternative efficient approaches to address these computational challenges, such as (1) initializing the calibration with random initial guesses, (2) multiple restarts of the LM algorithms throughout the minimization process, and (3) exploration of a series of alternative values for various parameters controlling LM performance (Lin et al., 2016). In addition, the compute speed deteriorates with the higher number of parameters used in calibration. It requires the computation of the Jacobian matrix and its pseudo-inverse, which can be computationally expensive for large-scale problems.

**2.6 Calibration Process, Parameters and Targets**

The calibration process in DVM-DOS-TEM is currently focused on the $C$ and $N$ annual cycles. Thus, calibrated parameters are associated with and adjusted to the major $C$ and $N$ fluxes and stocks in the vegetation and the soil. The calibration process follows a hierarchical approach (Figure 1), in which parameters to be calibrated are organized in hierarchical levels associated with (1) model complexity and feedback and (2) turnover of the processes the parameters are associated with. Therefore, parameters related to vegetation dynamics are calibrated first, followed by the slowest soil-related parameters.

The first step of the calibration relates to the simplest, fastest, first-order process in DVM-DOS-TEM, and consists of adjusting the rate limiting parameter of maximum C assimilation of the vegetation ($c_{max}$) driving vegetation GPP. Under baseline climate, the main limiting parameter of vegetation productivity in the Arctic is N availability (Chapin and Kedrowski, 1983). Therefore, $c_{max}$ is calibrated to reproduce estimates of GPP from fertilization experiments where N limitation is ignored (GPP*). When fertilization experiments are not available for the community/region of interest, GPP* is estimated by applying a multiplicative factor to observed GPP under natural conditions. This multiplicative factor is estimated from published fertilization experiments in similar communities and computed as the ratio

between GPP estimated in fertilized plots and GPP estimated in control plots. Based on the literature,
this fertilization factor can vary from 1.25 to 1.5 (Ruess et al., 1996; Shaver and Chapin, 1995).
The second step of the calibration process consists of turning on the representation of $N$ limitation on
vegetation productivity in the model (Euskirchen et al., 2009) and calibrating the rest of the vegetation-
related parameters. In the current workflow, it consists of three substeps. These substeps could follow a
different order based on the preference of the user and the specifics of a given site.  These are rate-
limiting parameters for maintenance respiration ($Kr_b$), maximum plant N uptake ($n_{max}$), C and N
litterfall ($c_{fall}$ and $n_{fall}$ respectively). These parameters are adjusted until DVM-DOS-TEM outputs
match observations of GPP and NPP, plant N uptake (Nup), and vegetation C and N pools,
respectively). Target values of these variables are listed in Table 1. It is important to note that the
parameters $Kr_b$, $c_{fall}$, and $n_{fall}$, as well as the variables for vegetation $C$ and $N$, are specified per PFT
and per compartment (leaf, stem, root).
In the third step, the rate-limiting parameters of soil heterotrophic respiration ($kdc$) and rate of
microbial $N$ uptake ($n_{micb}^{up}$ ) are calibrated as soil processes and takes longer to run in comparison to the
first two steps. These parameters are adjusted until DVM-DOS-TEM outputs match observations of soil
organic $C$ and available $N$ stocks. Target values of these variables are listed in Table 2. In a final state,
vegetation-related parameters are checked for a final adjustment after soil calibration, as soil processes
can feedback to vegetation dynamics.

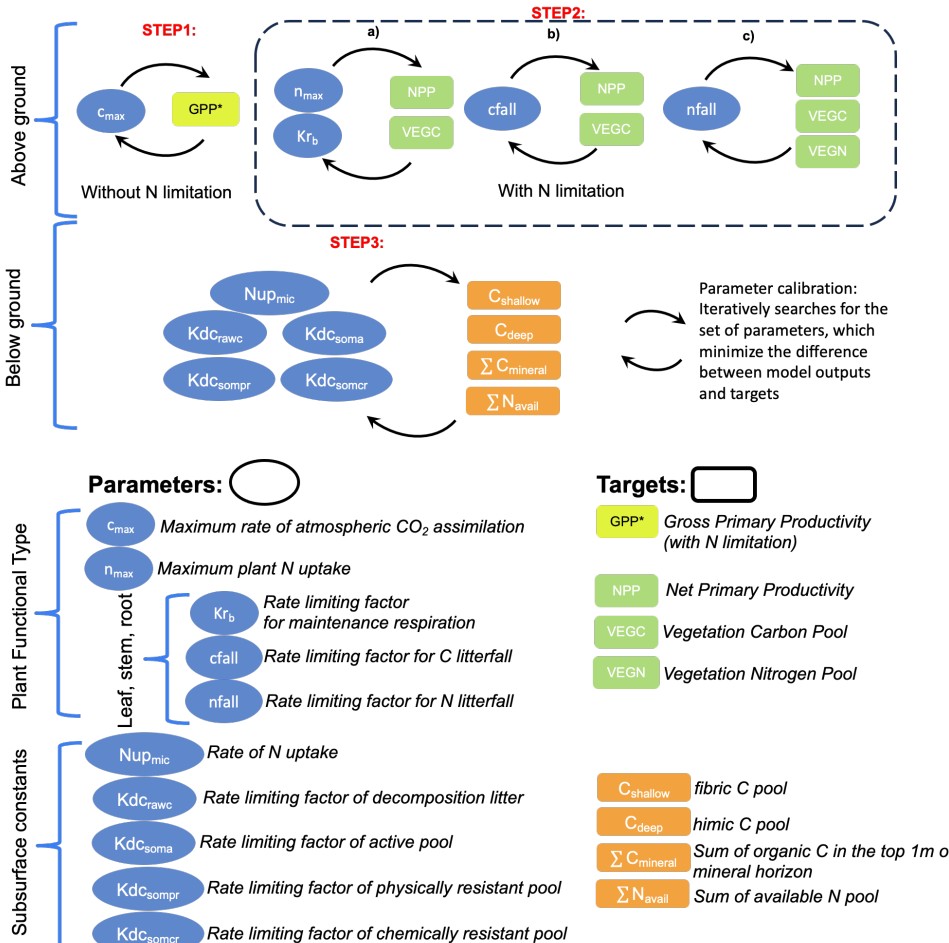

**Figure 1**. Schematics of the DVM-DOS-TEM model parameters and targets participated in the calibration process.

### 2.7 Calibrations setup and evaluation metric

Table 3 shows the parameter values used to calculate synthetic target values. We established four cases by perturbing the parameters by 10%, 20%, 50%, and 90% from their original values. For each case, the MADS calibration function randomly sampled ten sets of parameters within the specified ranges. These ten sets of randomly perturbed parameters were then optimized using the MADS algorithm. For each set of calibrated parameters and targets, we computed the root mean square error (RMSE) and relative error (RE) metrics. RMSE is employed to measure the magnitude of varying quantities, while RE gauges the absolute difference relative to the actual values. Given that some parameters are small (less than $10^{-3}$), the relative error provides more informative insights. The following equations were used to compute these metrics:

$$RMSE = \sqrt{(\overline{x} - x)^2} , \tag{1}$$

$$RE = |\frac{\overline{x}-x}{x}| \cdot 100\%, \tag{2}$$

where $\overline{x}$ is the mean of the best five out of ten computed target/parameter matches and $x$ is a synthetic target value.

To ensure the selection of the best-fitting parameters, we sorted error values from the lowest to the highest. Then, we selected the top five parameter sets, calculated their mean values, and compared these averaged parameters with the synthetic target values and known parameters.

**Table 3**: Synthetic parameter values for the black spruce forest site used in the parameter calibration process.

| Name | Parameters | Units | Plant Functional Types | | | |
|---|---|---|---|---|---|---|
| | | | Evergreen Tree | Deciduous Shrub | Deciduous Tree | Moss |
| Maximum rate of atmospheric $CO_2$ assimilation | $c_{max}$ | gC/m2/month | 381.19 | 113.93 | 210.48 | 93.31 |
| Maximum rate of plant N uptake | $n_{max}$ | gN/m$^2$/month | 3.38 | 1.55 | 1.0 | 3.55 |
| rate limiting factor for C litterfall for leaf | $c_{fall}^{leaf}$ | month$^{-1}$ | 0.0011 | 0.05 | 0.025 | 0.02 |
| … for stem | $c_{fall}^{stem}$ | month$^{-1}$ | 0.0034 | 0.0048 | 0.0036 | |
| … for root | $c_{fall}^{root}$ | month$^{-1}$ | 0.0052 | 0.0012 | 0.026 | |
| Rate limiting factor for N litterfall for leaf | $n_{fall}^{leaf}$ | month$^{-1}$ | 0.0102 | 0.045 | 0.018 | 0.007 |
| … for stem | $n_{fall}^{stem}$ | month$^{-1}$ | 0.001 | 0.001 | 0.005 | |
| … for root | $n_{fall}^{root}$ | month$^{-1}$ | 0.003 | 0.007 | 0.008 | |
| Rate limiting factor for maintenance respiration for leaf | $Kr_b^{leaf}$ | month$^{-1}$ | -6.0 | -3.45 | -2.95 | -4.65 |

| | | | | | | | |
|---|---|---|---|---|---|---|---|
| … for stem | $Kr_b^{stem}$ | month⁻¹ | -4.88 | -5.15 | -6.65 | | |
| … for root | $Kr_b^{root}$ | month⁻¹ | -8.2 | -6.2 | -3.2 | | |


**Table 4**: Synthetic below-ground target values for the black spruce forest site used in the parameter
calibration process

| Name | Parameters | Unit | Value |
|---|---|---|---|
| Rate of microbial N uptake | $n_{micb}^{up}$ | $gg^{-1}$ | 0.4495 |
| Rate limiting factor of litter decomposition | $kdc_{rawC}$ | $month^{-1}$ | 0.634 |
| Rate limiting factor of active pool decomposition | $kdc_{soma}$ | $month^{-1}$ | 0.54 |
| Rate limiting factor of physically resistant pool decomposition | $kdc_{sompr}$ | $month^{-1}$ | 0.002 |
| Rate limiting factor of chemically resistant pool decomposition | $kdc_{somcr}$ | $month^{-1}$ | 0.00007 |

**2.8 Application of the calibration method to observed target values**
After validating our calibration method with synthetic data, we applied it to observed at the Black
Spruce site. The observational dataset was compiled using a combination of in-situ measurements and
values from existing literature (Tables 5 and 6). Unlike synthetic targets, observed values inherently
carry uncertainty, which must be accounted for in the calibration process. The uncertainty range in the
observed targets varied from 27% to 40% (maximum coefficient of variation estimated from
observations reported in Melvin et al., 2015) influencing the final calibrated parameter estimates. After
calibrating parameters using observed means as targets, we sampled one thousand parameter sets
around the calibrated parameter set with a ±5% variation for all parameters excluding $c_{max}$. This
approach was implemented to increase the probability of achieving an optimal match with observations,
thereby allowing for a higher set of optimal parameter estimates. Additionally, this process enabled us
to evaluate the impact of calibrated soil parameters on vegetation-related target values, which were
calibrated over shorter time intervals.
**Table 5**: Observed vegetation target values at the black spruce forest site used in the parameter
calibration process. Standard deviations are indicated in parenthesis and estimated from field
measurements (n=15, Melvin et al., 2015).

| Above-ground Target Names | Notation | Units | Plant Functional Types | | | |
|---|---|---|---|---|---|---|
| | | | Evergreen Tree | Deciduous Shrub | Deciduous Tree | Moss |
| Gross Primary Productivity without nitrogen limitation | GPP* | [gC/m²/year] | 306.07 (±106) | 24.53 (±8.4) | 46.53 (±15.9) | 54.23 (±18.5) |
| Net Primary Productivity | NPP | [gC/m²/year] | 153.04 (±39) | 12.27 (±3.9) | 17.36 (±8.2) | 27.10 (±11.1) |
| Vegetation Carbon Leaf | $C_{leaf}$ | [gC/m²] | 293.76 (±100) | 15.13 (±5.4) | 9.06 (±2.4) | 180.85 (±93.3) |
| Vegetation Carbon Stem | $C_{stem}$ | [gC/m²] | 1796.32 (±706) | 100.16 (±37) | 333.75 (±185) | |
| Vegetation Carbon Root | $C_{root}$ | [gC/m²] | 404.48 (±177) | 15.07 (±6.4) | 44.8 (±15.9) | |
| Vegetation Nitrogen Leaf | $N_{leaf}$ | [gC/m²] | 6.35 (±3.5) | 0.72 (±0.14) | 0.7 (±0.2) | 1.61 (±0.8) |
| Vegetation Nitrogen Stem | $N_{stem}$ | [gC/m²] | 24.34 (±11.3) | 2.48 (±1) | 9.45 (±4.9) | |
| Vegetation Nitrogen Root | $N_{root}$ | [gC/m²] | 0.17 (±0.04) | 0.01 | 0.03 (±0.1) | |


**Table 6**: Observed below-ground target values at the black spruce forest site used in the parameter
calibration process. Standard deviations are indicated in parenthesis and estimated from field
measurements (n=15, Melvin et al., 2015).

| Below-ground Targets Names | Notation | Unit | Value |
|---|---|---|---|
| Carbon Shallow | $C_{shallow}$ | g/m2 | 782.73 (±216.7) |

| | | | |
|---|---|---|---|
| Carbon Deep | $C_{deep}$ | g/m2 | 3448.46 ($\pm$955) |
| Carbon Mineral Sum | $\sum C_{mineral}$ | g/m2 | 41665.0 ($\pm$10580) |
| Available Nitrogen Sum | $\sum N_{avail}$ | g/m2 | 0.76 ($\pm$0.24) |


## 3 Results

### 3.1 Vegetation Targets

Depending on the range of parameter variance, our analysis revealed varying levels of accuracy between known synthetic parameters and those determined using the MADS search approach. In general, the variance between calibrated and synthetic values grew higher with a higher degree of parameter perturbation. The averaged RMSE values for all four PFTs showed similar increases (Figure 2) with an exception for $C_{stem}(c_{fall})$ deciduous shrubs, which made the RMSE score for the 10% variance higher than the 20% variance (Figure 2a and 2b). That is why we introduced the RE metric, which shows that the departure between synthetic and calibrated parameters increases with increasing perturbation and is the smallest for the 10% variance (Figure 3a). Additional analyses to explore the detailed relationship between parameter variance and RMSE for specific cases are presented in the supplementary materials (Figures S2-S5).

### 3.2 Vegetation Parameters

The RMSE for parameters was highest for $Kr_b^{root}$ in the evergreen tree PFT (Figure 3). Overall, $Kr_b$ and $n_{max}$ parameters exhibited the worst recovery compared to other parameters based on the RMSE metric. Conversely, REs were highest for $c_{fall}$ deciduous shrubs and less for $Kr_b$ paramters. The RE indicated that smaller parameter values, such as $n_{fall}$, deviated more significantly from their synthetic values. Interestingly the RE score showed the same error range for 10% and 20% variance ranges, whereas RMSE showed that 10% variance has the smallest error.

### 3.3 Soil parameters

In general, the RMSE values for the sub-surface target parameters were relatively small but increased with a higher variance range (Figure 4). Notably, $C_{deep}$ and $\sum C_{mineral}$ exhibited high RMSE values of 3.34 and 9.12, respectively, for the 10% variance range (Figure 4a). Despite this, the soil parameters for 10% variance showed the best match, with RMSE values less than 0.01. The RE for targets revealed increasing deviations from the synthetic parameter values for $\sum N_{avail}$. The RE for parameters indicated

that $n_{micb}^{up}$, $kdc_{rawC}$ and $kdc_{soma}$ had higher deviations from their respective synthetic values for the
50% and 90% variance range, respectively.
**3.4 Comparison with Observations**
Figure 5 shows a comparison between observed and modeled target values after calibration. Both
observed and modeled values were normalized by dividing by the highest value within their respective
groups (e.g., GPP, NPP). The highest difference (exceeding 20% uncertainty) was observed for
Evergreen Trees (Black Spruce). Notably, we encountered challenges in accurately matching the values
of the $C_{stem}$ target and the values of $N_{stem}$ (Figure 5a). Additionally, while the calibration method
struggled to align the carbon in the soil mineral pool, it captured other soil target values (Figure 5a).
Overall, the results demonstrate that the calibration approach is effective and reliable for optimizing
DVM-DOS-TEM model parameters.

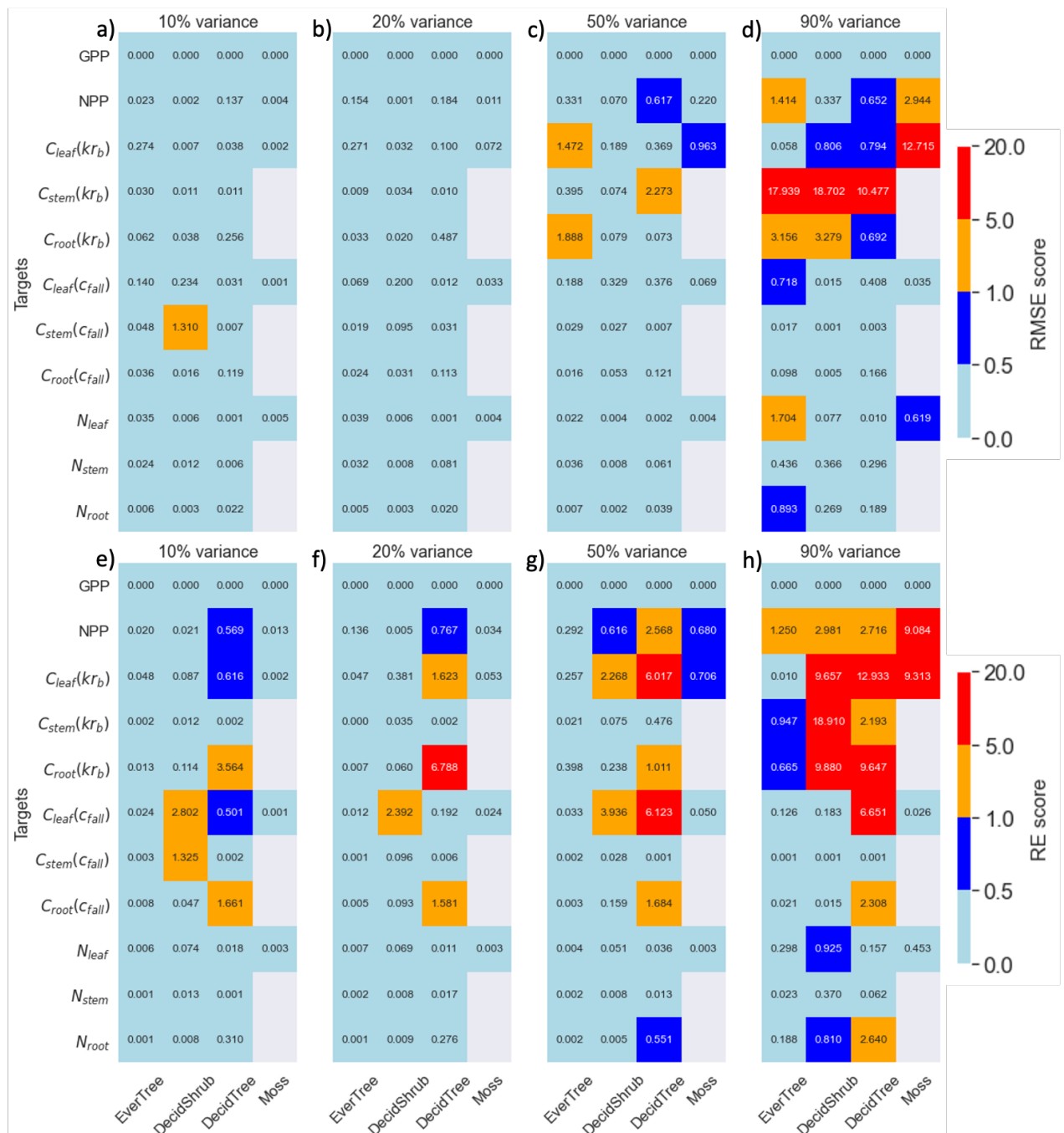

**Figure 2**. a), b), c), and d) are root mean square error (RMSE) metric and e), f), g), and h) are relative error (RE) metric for 10%, 20%, 50%, and 90% variance in the parameter range, correspondingly. Targets shown on y-axis, and plant functional types are on x-axis. The colorbar represents the RMSE and RE scores


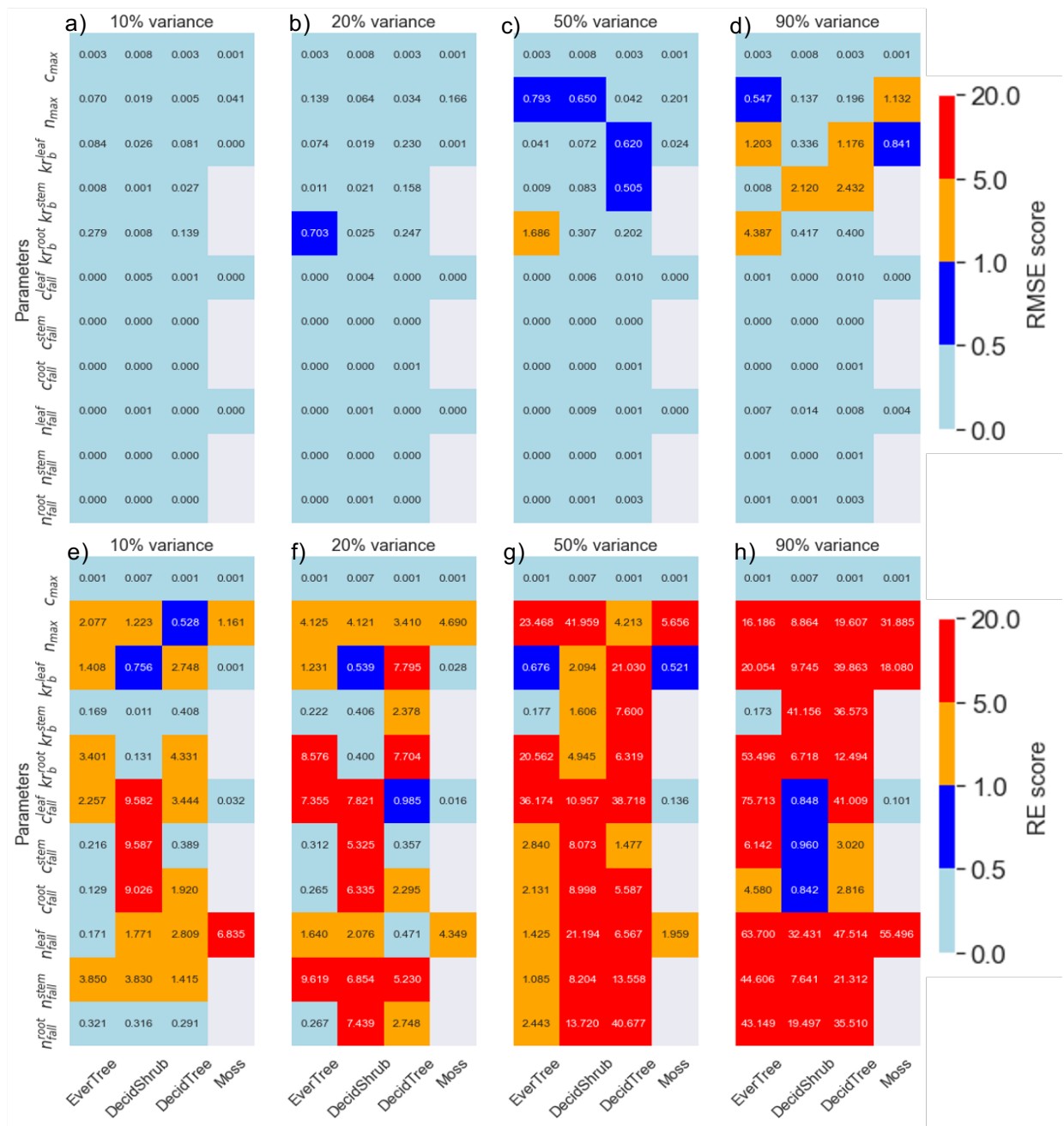

**Figure 3**. a), b), c), and d) are root mean square error (RMSE) metric and e), f), g), and h) are relative error (RE) metric for 10%, 20%, 50%, and 90% variance in the parameter range, correspondingly. DVM-DOS-TEM parameters shown on y-axis, and plant functional types are on x-axis. The colorbar represents the RMSE and RE scores.


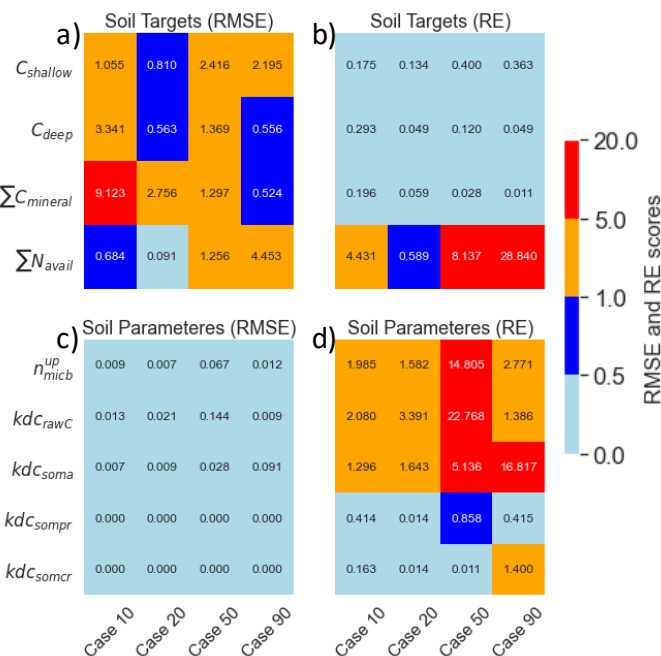

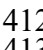

**Figure 4**. Comparison between calibrated and synthetic sub-surface target values (a) root mean square error (RMSE) and (b) relative error (RE) scores. Comparison between calibrated and synthetic sub-surface parameter values (a) root mean square error (RMSE) and (b) relative error (RE) scores for all range variances. The colorbar represents the RMSE and RE score.

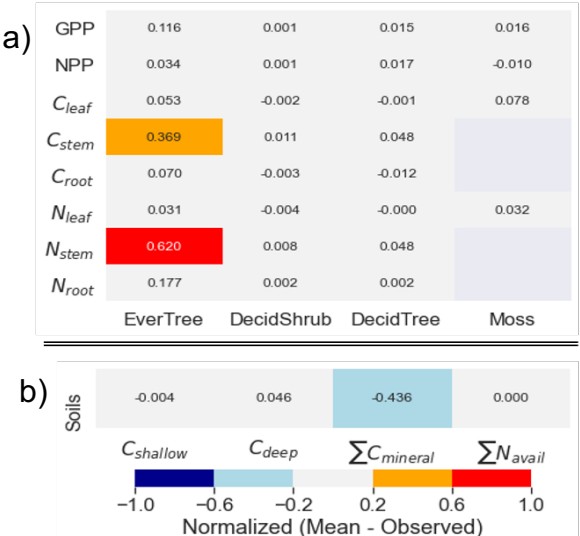

**Figure 5**.The comparison between observed and calibrated target values. The target values shown on y-axis, and plant functional types (a) and soil targets (b) on the x-axis. The colorbar represents the difference between normalized modeled and observed target values.

## 4 Discussion

Our findings highlight the challenges associated with calibrating carbon and nitrogen dynamics in high latitude permafrost ecosystems, particularly in accurately estimating carbon pools with slow turnover deep mineral soil carbon and allocation of partitioning carbon and nitrogen resources among within vegetation compartments to match in-situ observations closely. The strong interdependencies among parameters and state variables target values underscore the complexities of process-based modeling, reinforcing the need for automated calibration approaches like MADS to improve predictive accuracy.

### 4.1 Importance of the initial parameter guess

The initial parameter values, or initial guess, had minimal impact on the synthetic experiment, as the perturbed parameters were sufficiently close to the true values. However, for non-synthetic calibrations, the initial state is crucial, as starting with parameter values far from the true state can lead to non-convergence and significantly increase computation time(Nocedal and Wright, 2006). To address this, we developed parameter sensitivity methods to improve initial estimates (Briones et al., 2024). This approach utilized ensemble model simulations executed in parallel, systematically exploring parameter ranges through Latin hypercube sampling or uniform random sampling. By employing parallel processing before integrating parameters into the MADS calibration framework, we effectively refined initial estimates, minimized deviations from target values, and improved overall calibration efficiency.

### 4.2 Analysis of the recovery metrics

The mean parameter values calculated from the five best-matched MADS value predictions align closely with the synthetic parameter values, demonstrating the method's efficacy. The calculated REs for parameters indicate that the relative distance between the calibrated and the synthetic values increases with a higher parameter variance range, except RE for soil targets (Figure 4b, case 20%). For the soil targets, the RMSE for $\sum N_{avail}$ for 10% variance range were higher than 20% variance range. The higher RMSE for 10% variance than 20% variance range for vegetation-related targets as well as soil targets could be attributed to the limited number of cases (n=10) participated in each variance case. It is highly probable that increasing the total number of searches (higher than 10) would yield a more consistent pattern of decreasing accuracy with increasing variance.

### 4.3 Parameter-target relationship and small parameter values

The method demonstrated robust recovery of $c_{max}$ values, indicating that it performs best when there is a linear relationship between parameters and target values (Eq. S1). For parameters, which do not exhibit a linear relationship with their target values (e.g. $Kr_b$, Eq. S4), the calibrated parameters showed wider variance. Additionally, small parameter values, such as $n_{fall}$, corresponded to small range of sampled values, leading to insensitivity between $n_{fall}$ and vegetation $N$. To address this, we applied a logarithmic transformation to these and to some other small values for soil C rates.

### 4.4 The impact of $n_{max}$ on N uptake and NPP

Sensitivity between model parameters and targets is crucial for effective parameter calibration. We observed that the sensitivity between $n_{max}$ and $NPP$ was not strong (Eq. S2, Eq. S5), which led us to

combine its calibration with the $Kr_b$ parameter. Based on (Eq. S2), $n_{max}$ directly influences $N_{uptake}$.
An increase in $n_{max}$ enhances $N_{uptake}$, thereby increasing the total $N$ supply. Since $NPP$ is
proportional to $N_{supply}$ and inversely proportional to $N_{required}$, a higher $N$ supply can lead to a higher
$NPP$, provided that other factors remain constant. Therefore, despite the initial observation of weak
sensitivity, $n_{max}$ could have a considerable impact on $NPP$ due to its role in $N_{uptake}$ and the overall
$N_{supply}$. However, our target values for plant $N$ uptake are poorly constrained due to a lack of sufficient
observations. This underestimation of plant $N$ uptake could account for the observed lack of sensitivity
of NPP to $n_{max}$. This issue requires further investigation and currently underscores the importance of
accurately calibrating $n_{max}$ to ensure better simulation of ecosystem productivity.
**4.5 The Calibration Workflow**
Our findings indicate that calibrating one or two parameter sets at a time, while keeping other
parameters constant, is more effective than calibrating all parameters simultaneously. In the current
workflow, we combined $n_{max}$ and $Kr_b$ (Figure 1 Step a), which was based on the low sensitivity of
$n_{max}$ to $NPP$. Combining multiple variables in one calibration step increases the compute time and
could result in low match accuracy. On the other hand, sequential parameter calibration carries the risk
of losing accuracy for parameters calibrated in previous steps. To mitigate this risk, we include targets
from previous calibration steps in the current calibration step. For example, when optimizing for $n_{fall}$,
we include targets for $NPP$, vegetation $C$, and vegetation $N$.
Sequentially calibrating individual parameter sets is advantageous not only computationally but also in
preventing the occurrence of an underdetermined problem, which arise when the number of parameters
exceeds the number of targets. Undetermined problems exhibit a lower rate of convergence due to the
correlation between parameters and the sensitivity of multiple parameters to one or a few similar target
values. The study by Jafarov et al., (2020) showed that overdetermined problems with higher and
diverse number of target values, are more effective in recovering accurate parameter values.
**4.6 Sensitivity of the $Kr_b$ parameter to NPP and vegetation C**
The $Kr_b$ parameter exhibited higher sensitivity to both $NPP$ and vegetation $C$ compared to other
parameters. Despite the overall good model fitness, the deviation from the synthetic values for $Kr_b$ was
higher. This was primarily due to $Kr_b^{root}$ parameter for evergreen trees (Figure S3C) persistently
showed higher discrepancy. Its sensitivity can be explained by examining its role in the equations
governing maintenance respiration ($R_m$ Eq. S3). The relationship between biomass and maintenance
respiration is non-linear; $R_m$ increases as biomass increases, where $Kr_b$ controls the intercept of this
relationship (Tian et al., 1999). Since $NPP$ is computed as a resultant of $GPP$ and autotrophic
respiration, including $R_m$, any alteration in $Kr_b$ impacts $NPP$ directly (Eq. S9). This sensitivity
underscores the importance of accurately calibrating $Kr_b$ to ensure the correct simulation of ecosystem
productivity and C dynamics in the DVM-DOS-TEM.
**4.7 Vegetation and Below-Ground C stocks equilibrium time**
Due faster turnover, vegetation C and N stocks and fluxes equilibrate faster than soil C and N stocks
and fluxes. Thus we used a two-phase equilibration approach: 200 years for the vegetation and 2000
years for the soil. However, the C stocks achieved after 200 years of equilibration for vegetation might
shift when the model is run for an additional 1800 years to equilibrate soil. To mitigate this issue, we
developed equilibrium checks to ensure that the vegetation stocks remain stable and close to their
equilibrium values throughout the extended simulation period required for soil stocks equilibration.
These checks help identify significant departures from the initial equilibrium values of vegetation C and
N while allowing the model to run for a longer duration to achieve below-ground equilibrium. This
approach ensures the accuracy and stability of both vegetation and below-ground C and N stocks in
long-term model simulations.
Reversing the calibration sequence and starting from soil parameters is not only impractical in the
context of our model, but also computationally inefficient. Vegetation-related parameters are calibrated
first because vegetation carbon pools reach equilibrium significantly faster than soil carbon pools
whereas soil pools require longer timescales to stabilize. Beginning with soil parameters would thus
introduce unnecessary complexity and substantially increase the total computational cost of the
calibration process. In addition, while the choice of calibration sequence may lead to slight variations in
the final parameter estimates, our results demonstrate that the proposed "hierarchical approach"
(breaking the parameter sets into smaller subsets) effectively recovers parameter values, even when for
90% parameter range variance. As we showed in this study, well-calibrated parameters exhibit a narrow
range of uncertainty, reinforcing the robustness of the method.

**4.8 Observed target values**
The results of parameter calibration using site-specific observations indicate challenges in accurately
matching $C_{stem}$ and $N_{stem}$ target values for the evergreen plant functional type. This discrepancy could
be related to the allocation scheme of the model, attributing NPP resources to the various compartments
of the plant (Fox et al., 2018). Additionally, the model struggled to maintain the assigned carbon value
for $\sum C_{mineral}$. The difficulty in calibrating $C_{stem(E)}$ and $C_{root(E)}$ for evergreen trees can be partially
attributed to strong parameter interdependencies (see Figures SI7–SI10). For instance, $Kr_b^{leaf(E)}$
exhibits simultaneous correlations with both $C_{stem(E)}$ and $C_{root(E)}$ (Figure S7), while $c_{fall}^{stem(E)}$ shows an
inverse correlation with N leaf, stem, and root (Figure S8). These multi-target dependencies introduce
additional complexity, making it challenging to achieve a precise match for individual target values.
Similarly, the $\sum C_{mineral}$ target value is strongly influenced by $kdc_{soma}$ and $kdc_{sompr}$, both of which
exert substantial control over $C_{deep}$ and $\sum N_{avail}$ target values. These interactions underscore the
systemic constraints imposed by parameter interdependencies. Furthermore, this discrepancy could be
related to the functions controlling vertical transfers of carbon between horizons and the vertical
distribution of carbon quality (Harden et al., 2012). The model consistently showed that longer
equilibration times lead to a reduction in the mineral soil carbon pool. This was also observed by
Schaefer and Jafarov, (2016) in a different process-based ecosystem model, where they addressed the
issue by incorporating substrate availability constraints to prevent long-term carbon loss. Given the
complexity of these interdependencies, further investigation is needed, though it falls beyond the scope
of this study.
The calibration of rate-limiting soil parameters that influence C and N stocks and turnover directly
impacts vegetation productivity by modulating nitrogen availability. Figure S10 shows a significant
correlation between microbial nitrogen uptake and $C_{leaf(DS)}$ of deciduous shrub, highlighting the
interaction between soil processes and vegetation-related parameters. While long-term soil parameter
calibration inherently feedbacks into vegetation dynamics, the most substantial changes in vegetation-
related parameters typically occur during short-term model runs, resulting in minimal net changes over
extended simulations.

## 4.9 Limitations
There are cases where the model fails to accurately match target values due to poor data quality or its
inability to fully represent certain ecological processes (Dietze et al., 2018; Luo et al., 2016). Large
discrepancies between observed and modeled targets can hinder the convergence of the LM method,
requiring more iterations and leading to suboptimal agreement with observations. As previously
mentioned, starting with well-constrained initial parameter estimates can mitigate this issue, which can
be achieved by performing sensitivity analyses to identify the most influential parameters and refine
their ranges prior to calibration (Efstratiadis and Koutsoyiannis, 2010).
Additionally, calibrating soil-related parameters is computationally demanding, often resulting in a
substantial slowdown of the overall calibration workflow. Machine learning (ML) models offer a
promising solution by acting as surrogate models to approximate the equilibrium state, thereby reducing
the computational burden (Fer et al., 2018; Reichstein et al., 2019). However, implementing such
approaches necessitates large training datasets, often requiring thousands of model simulations to
achieve reliable predictions. Future research should explore the integration of ML-based calibration
techniques into the workflow, which could significantly enhance computational efficiency and further
improve model accuracy (Castelletti et al., 2012; Dagon et al., 2020).

## 5. Conclusion

In this study, we showed that the developed MADS parameter calibration method for the DVM-DOS-
TEM can effectively recover the synthetic parameter set, optimizing labor and time, and enhancing
reproducibility of the calibration process. By implementing a structured workflow that calibrates one or
two parameters at a time and including equilibrium checks the method ensured accurate parameter
estimation even for high variance parameter range. The primary advantage of the semi-automated
MADS calibration approach is its significant enhancement of repeatability and clear quantification of
calibration performance. In contrast, manual calibration processes are often difficult to reproduce as it is
impractical if not impossible, to record users continuous adjustments to parameters values until
improved results are achieved. Additionally, appreciation of model improvement by the user is often
subjective as running a statistical evaluation at each parameter adjustment would be too time
consuming. In the approach demonstrated in this study, we introduced a calibration metric that provides
a quantifiable measure of the overall quality of the calibration. This metric enhances reproducibility by
allowing future users working on the same site to follow the established workflow and reliably
reproduce the calibrated parameter and target values. The RMSE quantifies the average differences
between calibrated and observed (synthetic) values, while the RE metric indicates deviations from the
synthetic values.
In all calibration experiments, we utilized only ten randomly perturbed initial parameter sets within a
specified variance range. Our results indicated that perturbation ranges of 10%-20% were equally
effective in achieving optimal target/parameter calibration. However, increasing the number of random
perturbations could potentially shift the statistics, favoring a 10% variance range.
While the choice of the initial guess is crucial, its impact was mitigated in our study due to the design
involving variance around synthetic parameter values. The developed method significantly reduces the
labor and time required for calibrating DVM-DOS-TEM model parameters. However, it does not
entirely replace the need for human intervention. Users still need to understand the specifics of the
model and the relationship between parameters and targets, as well as conduct post-processing
assessments of the fit. In future work, we will apply this method to data processed at multiple study
sites to validate further and refine the calibration approach.
The application of the calibration method to site-specific observations revealed challenges in accurately
matching $C_{stem}$, $N_{stem}$ and $\sum C_{mineral}$ values, primarily due to parameter interdependencies and data
uncertainties. Discrepancies between observed and modeled target values exceeded the known the
measurement uncertainty, suggesting that structural uncertainty within the model may contribute to
these deviations. This indicates a potential need for a more detailed representation of ecological
processes to improve model accuracy. However, these challenges may be site-specific and may not
necessarily apply to other ecosystem types. Despite these limitations, the study demonstrates the
effectiveness and reliability of the calibration approach while identifying key areas for future model
refinement.
**6. Data and model availability**
The version of the model used in these simulations, along with the calibration scripts, auxiliary files
(including plots presented in the paper), and corresponding output files, is available in Jafarov, (2024).
**7. Author contributions**
EEJ designed and executed the experiment. HG supervised in experiment design. VV supervised with
MADS model. RR, TC, and DT provided technical support on the DVM-DOS-TEM model. VB, AK,
ALM, BM, C-CC, and JC tested calibration approach. TS technical support on scientific computing. All
authors participated in manuscript writing and editing. SMN and BMR provided overall supervision and
research funding.
**8. Competing interests**
The contact author has declared that none of the authors has any competing interests.
**9. Disclaimer**

## 10. Acknowledgments

This work was supported by the Quadrature Climate Foundation (grant number 01-21-000094) and catalyzed through the Audacious Project (Permafrost Pathways) to SMN and BMR.

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
