# Peer review of "Estimation of above- and below-ground ecosystem parameters for"

_Geoscientific Model Development, 2024_

## Author Response (AR1)

We sincerely thank the reviewers for their time and effort in providing valuable feedback to enhance this manuscript. Below, we address the reviewers' comments in detail.

**RC1**

The manuscript by Jafarov et al. present an automated parameter calibration method applied to the MADS package with the aim to improve the calibration of a set of carbon dioxide and nitrogen rates parameters in high latitude regions. The method presented uses an initial guess as a seed for the calibration, and discusses the accuracy of the parametric calibration based on four cases of parameter perturbation by 10%,20%,50% and 90%. The accuracy of the obtained calibration is then assessed by comparison with the target values.

In my opinion, this manuscript presents an interesting and relevant approach to parameter calibration, that can be widely used in panarctic environments, where the site information is scarce (particularly in winter) and the sensitivity of models to certain parameters is very high. The approach to the problem is clear as well as the methodology used and the steps followed to obtain the results. The conclusions are well supported by the results. The abstract reflects the content of the paper and the title is also adequate to the content.

I do have however, one major comment to the approach used. In the manuscript is underlined the importance of calibrating the model against data. Moreover, in the methods section is specified that the observation data is available in the selected site for the calibration. Despite this, the authors choose to calibrate against synthetic data. I can imagine the reason to do this is to simplify the perturbation parametrization, but as the manuscript underlines the importance of the first guess in the calibration convergence, I wonder if this is done to guarantee the results convergence. In any case, due its obvious importance, I think is necessary to run the calibration method against the available observed data. If the results are convergent, its comparison with the synthetic data calibration, and how differ from the differences between the synthetic and the observed data itself would be a valuable addition to the manuscript. On the contrary, if the calibration with observed data is divergent for all perturbations, that would limit the applicability of the method. In case of a convergence for at least some of the perturbations, would give valuable information about the applicable range of perturbation, and necessary first guess accuracy range. In any case, I believe that in order to discuss the value of the method presented, the calibration with observed data is necessary.

The primary objective of this paper was to provide a proof-of-concept for the MADS method and demonstrate its robustness, even when calibrated parameters span a wide range. Initially, we intended to present a more refined calibration methodology in a follow-up study. However, after careful consideration, we decided to extend our analysis to include parameter calibration using observed targets.

To reflect this addition, we incorporated Subsection 2.8, which highlights the complexities of calibrating with real-world data and details the techniques employed to optimize computational efficiency. Subsection 3.4 was added to the Results section to present the outcomes of this calibration, along with a corresponding discussion subsection 4.8. Additionally, we included tables 5 and 6 of observed target values in the manuscript and added several correlation plots (Figures S7-S10) to the Supplementary Information (SI) to support the discussion.

In addition, I have some individual minor comments:

On the subsection 2.1 Synthetic data for Black Spruce forest site, the setup is presented as a forest community type (CMT). I assumed this CMT is composed by 4 plant functional types (PFT) and one "soil" type based in the information on table 1 and 2. This classification is later introduced in 2.2, lines 166 to 170. For clarity, I would recommend move this paragraph or a version of it from 2.2 to 2.1, before the tables reference.
We moved subsection 2.2 after the paragraph describing the Murphy Dome site. Now, Tables 1 and 2 are introduced after the model description.

On the caption of Figure 1 a reference is made to the website of MADS, I think it would be better to move this link to the data and model availability section (6)
We moved Figure 1 into SI.

On line 242, the acronym definition for the Levenberg-Marquardt algorithm is not necessary as it was defined on line 223.
The URL has been removed and is now in the references.  The duplicate acronym has been removed.

On the results section, the first results referenced, in line 304, are the results located on the supplementary, while the first results in the manuscript are presented in line 306. My understanding is that the supplementary materials should be provided to support the information on the manuscript but should not be an integral and necessary part of it. I can understand that the addition to the main manuscript of the figures S2 to S5 in the main manuscript would extend it significantly and unnecessarily, as the information is an extension of the results presented in figures 3,4 and 5. I would recommend to reorganize the text, so the supplementary figures are presented after the figures on the main manuscript, and as an additional source of information, not a result per se.
We agree. We changed the reference from Figure S2-S5 to Figure 3 and added a sentence at the end of the paragraph in Section 3.1: "Additional analyses to

explore the detailed relationship between parameter variance and RMSE for specific cases are presented in the supplementary materials (Figures S2-S5)."

Figures 3, 4 and 5 present the same colour range for widely different value ranges. In my opinion this is confusing as, comparing between % variance results, would seem visually that higher values on a higher variance are lower than lover values in a lower variance (i.e. Figure 3: Cleaf 20% variance column 1 is lower than the same parameter values for 90% variance in columns 2 and 3). I would recommend use a unified colour scale, so the values between variances are clearly inter-comparable. As the ranges between the colour scales depending the variance are highly variable, I would recommend use value-separated colour ranges for the error score (i.e. for figure 3a-d: 0-1 blue, 1-5 orange, 5-20 red).
Great suggestion. We updated all figures as suggested.

**RC2**

The authors developed an automated parameter calibration method using a software MADS to optimize the DVM-DOS-TEM model for permafrost ecosystems. MADS employed a LM optimization algorithm to enhance the efficiency of exploring parameter combinations and identifying the most optimal sets. The method was tested using synthetic data from a black spruce boreal forest to demonstrate its effectiveness in enhancing model calibration. The analytical pipeline appears to be expertly executed, but the results are relatively simple and fail to clearly showcase the method's advancements over other calibration approaches outlined in the introduction. Also, the structure of the paper requires considerable improvement to enhance its coherence. I have noted a few questions and concerns for the authors to address to help clarify the paper.

The hierarchical approach described in Figure 2 seems logical, with a clear progression from vegetation dynamics to soil-related parameters. However, the explanation of how the calibration process proceeds in a hierarchical structure is not fully clear. Can the hierarchical approach be visualized more explicitly to clarify how do changes in one parameter influence others, especially when calibrating slower soil-related parameters after vegetation-related ones?

The visualization presented in Fig. 2, now Fig. 1, illustrates the conceptual framework of our calibration process. Step 1 is an independent prerequisite that must be completed before calibrating any other parameters. Step 2 allows for considerable flexibility in approach. For instance, part (a) can be subdivided to calibrate nmax and krb separately, and the sequence of substeps can be adjusted (e.g., (c), (b), (a) or (a), (c), (b)). Our chosen sequence was informed by prior experience in calibrating model parameters.

In any case, the increase in the complexity and parameter number leads to the conditions when parameters influence one another. There are cases when parameters are correlated with one another and exert high influence on more than one target value. In instances where close match is not possible due to their coupled sensitivity to parameter change. In these cases, one can choose the target with the higher value or higher priority.

To address how calibration of slower soil-related parameters impacts vegetation-related match after final calibration, we added parameter calibration using observed targets.
In subsection 2.8, we highlight the complexities of calibrating with real-world data and detail the techniques employed to optimize computational efficiency. Subsection 3.4 was added to the Results section to present the calibration outcomes, along with a corresponding discussion subsection 4.8. Additionally, we included tables 5 and 6 of observed target values in the manuscript and added several correlation plots (Figures S7-S10) to the Supplementary Information (SI) to support the discussion.

The soil-related parameters, such as heterotrophic respiration and microbial N uptake, are calibrated in the third step appears to involve significant adjustments based on soil C and N stocks, but it would be useful to understand how feedback from soil processes is incorporated into the vegetation-related parameter calibration.

The calibration of the rate limiting parameters influencing soil carbon and nitrogen stocks and turnover feedback to vegetation productivity by impacting nitrogen limitation. The following text was added to the Discussion subsection 4.8

"The calibration of rate-limiting soil parameters that influence C and N stocks and turnover directly impacts vegetation productivity by modulating nitrogen availability. Figure S10 shows a significant correlation between microbial nitrogen uptake and $C_{leaf(DS)}$ of deciduous shrub, highlighting the interaction between soil processes and vegetation-related parameters. While long-term soil parameter calibration inherently feedbacks into vegetation dynamics, the most substantial changes in vegetation-related parameters typically occur during short-term model runs, resulting in minimal net changes over extended simulations."

Notably, maintaining $C_{mineral}$ close to the target value proved to be more challenging, an issue we have addressed in subsection 4.8.

Calibrating soil-related parameters is computationally demanding. It would be valuable to discuss how this increased computational complexity affects the overall calibration workflow. Are there any strategies in place to optimize the calibration process or reduce computation time?

Calibrating soil-related parameters is indeed computationally intensive, which can pose significant challenges for the overall calibration workflow. While optimizing computation time for

these parameters is an important avenue for improving the calibration process, it falls outside the scope of the current study. We added subsection 4.9 to the Discussion section.

"Additionally, calibrating soil-related parameters is computationally demanding, often resulting in a substantial slowdown of the overall calibration workflow. Machine learning (ML) models offer a promising solution by acting as surrogate models to approximate the equilibrium state, thereby reducing the computational burden (Fer et al., 2018; Reichstein et al., 2019). However, implementing such approaches necessitates large training datasets, often requiring thousands of model simulations to achieve reliable predictions. Future research should explore the integration of ML-based calibration techniques into the workflow, which could significantly enhance computational efficiency and further improve model accuracy (Castelletti et al., 2012; Dagon et al., 2020)."

The results showed that the RMSE and RE metrics were not always in agreement, especially for parameters like nfall and Krbroot. How the model handles different types of error across parameter scales?

The discrepancy between RMSE and RE metrics, particularly for nfall and Krbroot, underscores how the model handles different types of error across parameter scales. In the synthetic case, where true parameter values were available, we found it informative to use both metrics to assess calibration performance.

The selection of RMSE and RE was deliberate, as these metrics offer complementary perspectives: RMSE quantifies the average deviation across all data points, whereas RE captures the relative distance between synthetic and calibrated parameters. The choice of which metric to prioritize depends on the specific analytical objective.

Given that calibration errors with observed data were substantially higher, we found it more informative to compare normalized differences, as illustrated in the newly added Figure 5.

The results focus on the calibration and recovery of parameters but does not provide enough detail on the model's overall evaluation against independent data. How does the calibrated model perform in simulating ecosystem carbon and nitrogen cycle? Is there any validation step against out-of-sample data? The lack of how do these methods compare in terms of computational time, accuracy, and robustness to non-linear relationships.

We added parameter calibration using observed targets see our first reply.

The discussion does not mention potential limitations or challenges of the calibration process itself. What happens if the model fails to match target values due to poor data quality or the inability to represent certain ecological processes adequately? A more critical reflection on the potential weaknesses of the calibration process would help improve transparency and guide future improvements to the model.

Great suggestion. We added the Limitation Subsection to the Discussion.

Minor comments:

The structure of the paper needs refinement. Many short, two- to three-sentence paragraphs disrupt the flow of ideas, making the content feel fragmented. Combining related ideas into well-structured paragraphs would significantly improve readability and coherence.

We have refined the structure of the manuscript to enhance readability and coherence by merging related ideas into well-structured paragraphs where appropriate.

The opening introduction does not clearly explain the extent to which parameter uncertainty plays a role in shaping model predictions or how important it is compared to other sources of uncertainty.

We have revised the introduction to clearly articulate the role of parameter uncertainty in shaping model predictions and its relative importance compared to other sources of uncertainty. The updated section now explicitly discusses how parameter uncertainty influences model outcomes and interacts with structural and input data uncertainties to affect prediction accuracy.

As for Figure 1, it doesn't add much value to the paper. It simply illustrates the use of a function already documented in the software's technical guide. Removing this figure would eliminate unnecessary clutter.
We removed Figure 1 and the text describing it from the manuscript and added it to the SI.

Lines 392-400: Are there any figures or tables to demonstrate these results
We added calibration with actual observations.

---

## Author Response (AR2)

We sincerely appreciate the time and effort the reviewer has dedicated to evaluating our manuscript. While we respect the review process, we respectfully disagree with the three main points raised in the evaluation that led to the rejection decision. We believe that our responses below provide further clarification to address any misunderstandings. Therefore, we kindly request that the decision be reconsidered.

**Reviewer Comment:**

"The study calibrates parameters using synthetic data, which is generated from the same model being calibrated. Even if the proposed method can successfully recover the set of parameters, I don't see the value in this analysis for advancing our understanding of permafrost ecosystem dynamics."

**Response:**

As stated in Lines 152–159 of the introduction, our study is methodological in nature, aiming to enhance the efficiency of the calibration process in a complex ecological model. This is the first time that the MADS algorithm has been applied to calibrate parameters in a terrestrial ecosystem model. Our primary objective was to demonstrate the effectiveness of the method itself, and the use of synthetic data is a standard and widely accepted approach in many scientific disciplines for this purpose. Synthetic data provides a controlled environment to validate calibration techniques before applying them to real-world data, where additional uncertainties and biases exist.

Furthermore, in response to the reviewer's request, we incorporated a calibration using real observations. This additional analysis demonstrates satisfactory calibration for over 90% of the target values (see Figure 5). As outlined in the results, this quantification confirms the validity of our calibration approach against real observations.

Finally, the objective of our manuscript was not to address potential structural issues that the calibration results may have highlighted. Instead, our approach improves calibration efficiency, enabling faster and more effective calibrations. This, in turn, allows for a broader application across diverse ecosystems, enhancing the representation of spatial heterogeneity in boreal and Arctic landscapes.

**Reviewer Comment:**

"If the model has structural errors that misrepresent a process, calibrating based on a biased or erroneous simulation would be meaningless."

**Response:**

While all models are simplifications of natural processes, this does not inherently mean they contain structural errors. Additionally, observational data itself carries significant uncertainties, as highlighted in our manuscript.

Calibration remains a valuable process because it helps identify areas where improvements are needed, both in data collection and in the mathematical representation of physical processes. Even if a model contains some level of structural uncertainty, calibration provides critical insights into parameter sensitivity and model behavior, helping to refine future versions of the model.

**Reviewer Comment:**

"However, if the authors can demonstrate that the method performs well on 'ground truth' data derived from in-situ measurements and remote sensing, I would find it more valuable. But according to the comparison between observed and calibrated results (Figure 5), the performance seems not ideal. Overall, I think this study is going in the wrong direction."

**Response:**

We strongly disagree with this assessment. Our study is moving in the right direction precisely because we openly acknowledge and address model-data mismatches rather than obscuring them. Furthermore, our calibration shows satisfactory results for more than 90% of our target variables, showcasing the validity of our method.

Moreover, as we discuss in our manuscript, the calibration performance may vary depending on site-specific conditions. The observed mismatches in Figure 5 do not indicate a failure of the methodology but rather highlight the inherent challenges of working with real-world data. Such challenges reinforce the need for improved calibration techniques, which is exactly what our study aims to contribute to.

**Reviewer Comment:**

"The authors employ a hierarchical approach to parameter calibration, but the rationale for the chosen order (e.g., calibrating vegetation parameters before soil parameters) is not clearly justified. Would reversing the calibration sequence affect the results? How sensitive is the method to the order of parameter calibration? Addressing these questions is critical for evaluating the robustness and generalizability of the proposed approach."

**Response:**

We addressed this concern in our initial response. To reiterate, the first step of our hierarchical calibration approach is required and consistent across applications and works well in most cases. The sequence of subsequent steps (Step 2 onward) is flexible and can be adjusted by the user. The key principle behind our approach is that calibrating all parameters simultaneously often

leads to poor results due to an excessive number of degrees of freedom. By breaking the calibration into smaller, sequential steps, we improve convergence and parameter estimation. We refer the reviewer to Section 4.7 and the final sentence of Section 4.8:

"While long-term soil parameter calibration inherently influences vegetation dynamics, the most significant changes in vegetation-related parameters typically occur during short-term model runs, resulting in minimal net changes over extended simulations."

To further clarify our rationale, we have added the following text to Section 4.7.

"Reversing the calibration sequence and starting from soil parameters is not only impractical in the context of our model, but also computationally inefficient. Vegetation-related parameters are calibrated first because vegetation carbon pools reach equilibrium significantly faster than soil carbon pools whereas soil pools require longer timescales to stabilize. Beginning with soil parameters would thus introduce unnecessary complexity and substantially increase the total computational cost of the calibration process. In addition, while the choice of calibration sequence may lead to slight variations in the final parameter estimates, our results demonstrate that the proposed "hierarchical approach" (breaking the parameter sets into smaller subsets) effectively recovers parameter values, even when for 90% parameter range variance. As we showed in this study, well-calibrated parameters exhibit a narrow range of uncertainty, reinforcing the robustness of the method."

**Reviewer Comment :**

"The manuscript acknowledges the challenge of equifinality but does not present a clear strategy to address it. The results indicate that the choice of initial parameter values has a greater impact on calibration outcomes when using observational data compared to synthetic data. This suggests that the proposed approach may be less reliable when applied to real-world ecosystems, where uncertainties and biases in observations are unavoidable. The study would benefit from a more rigorous discussion of how to mitigate equifinality and improve the method's performance on real-world datasets."

**Response:**

Equifinality is a well-documented mathematical challenge and a fundamental issue in many modeling applications, particularly in environmental and ecological modeling. It falls within the class of ill-posed problems, where multiple parameter sets can yield similar model outputs. While we do not claim to solve equifinality in our study, we explicitly acknowledge its presence and employ established techniques, such as parameter perturbation and multiple calibration runs to assess its impact.

Our approach involves conducting multiple calibration tests and evaluating convergence. If repeated calibrations converge toward a consistent parameter set, we gain confidence in the

results. Respectfully, the reviewer's expectation that equifinality can be fully eliminated suggests a misunderstanding of the inherent limitations of calibration in complex environmental models.

In summary, the statement that our method is unreliable in real-world ecosystems overlooks the fact that all models operate under uncertainty. Our study does not claim to eliminate uncertainty but rather provides a structured approach to improving parameter estimation.